# CONTEXT AUTOENCODER FOR SELF-SUPERVISED REPRESENTATION LEARNING

## ABSTRACT

We present a novel masked image modeling (MIM) approach, context autoencoder (CAE), for self-supervised representation pretraining. The goal is to pretrain an encoder by solving the pretext task: estimate the masked patches from the visible patches in an image. Our approach first feeds the visible patches into the encoder, extracting the representations. Then, we make predictions from visible patches to masked patches in *the encoded representation space*. We introduce an alignment constraint, encouraging that the representations for masked patches, predicted from the encoded representations of visible patches, are aligned with the masked patch presentations computed from the encoder. In other words, the predicted representations are expected to lie in the encoded representation space, which empirically shows the benefit to representation learning. Last, the predicted masked patch representations are mapped to the targets of the pretext task through a decoder.

One additional characteristic is that our approach encourages the separation of the representation learning part (encoder), and the pretext task completion part that will be replaced by the downstream task part. In contrast, previous MIM methods (e.g., BEiT and MAE) couple the two parts, potentially limiting the representation learning quality. We demonstrate the effectiveness of our CAE through superior transfer performance in downstream tasks: semantic segmentation, and object detection and instance segmentation.

## 1 INTRODUCTION

We study the masked image modeling task for self-supervised representation learning. Masked image modeling (MIM) is a pretext task of masking some patches of the input image and estimate the masked patches from the visible patches. It is expected that the resulting encoder pretrained through solving the MIM task is able to extract the patch representations taking on semantics that are transferred to solving downstream tasks.

BEiT (Bao et al., 2021) and the method studied in the ViT paper (Dosovitskiy et al., 2021), two MIM methods, learn a ViT to estimate the patch tokens and the pixels, respectively, and use the resulting ViT as the pretrained encoder. They take the visible patches and mask tokens representing the masked patches as input, and make estimations for both the visible and masked patches, where the estimations only for masked patches are evaluated during training. The two methods use the single ViT structure simultaneously for both representation learning and task completion. Thus, only the partial capacity of the ViT is explored for representation learning, limiting the representation quality. Masked autoencoder (MAE) (He et al., 2022) prepends an extra ViT structure that only receives visible patches as the so-called encoder followed by a lightweight decoder taking all the patches as input. Unfortunately, the decoder might play a partial role in representation learning, thus distracting the responsibility of representation learning.

We present a context autoencoder (CAE) approach, illustrated in Figure 1, for improving the encoding quality. We randomly partition the image into two sets of patches: visible patches and masked patches. The architecture contains an encoder, a latent contextual regressor with an alignment constraint, and a decoder, The encoder takes only the visible patches as input and learns the representations only for the visible patches. The latent contextual regressor predicts the masked patch representations according to the visible patch representations, where the predicted masked patch representations are

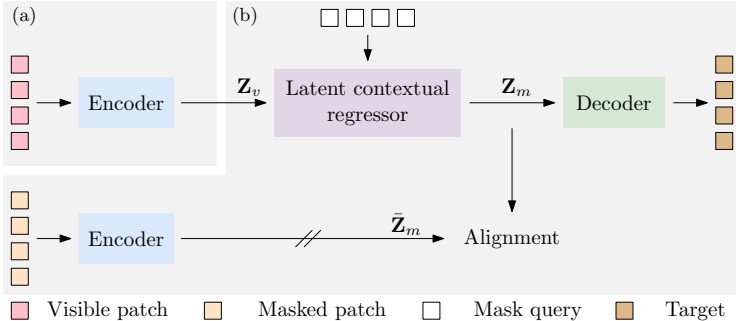

Figure 1: The pipeline of context autoencoder. Our approach feeds visible patches into the encoder and extracts their representations $\mathbf{Z}_v$ and then completes the pretext task by predicting the representations $\mathbf{Z}_m$ of the masked patches from the visible patches in the encoded representation space with latent contextual regressor and alignment constraint, and mapping predicted representations $\mathbf{Z}_m$ of masked patches to the targets. The pretrained encoder in (a) is applied to downstream tasks by simply replacing the pretext task part (b) with the downstream task part.

constrained to align with the masked patch representations computed from the encoder. The decoder maps the predicted masked patch representations to the targets for masked patches.

The prediction from the visible patches to the masked patches, i.e., generating a plausible semantic guess for the masked patches, is performed on the encoded representation space using latent contextual regressor. The predicted representations for the masked patches are constrained to match with the representations computed from the encoder, rendering that the predicted representations also lie in the encoded representation space. Making predictions in the encoded representation space encourages that the encoded representations take on a larger extent of semantics, empirically validated by the experiments.

In addition, the encoder in the top stream in Figure 1 operates on visible patches, only focusing on learning semantic representations. The CAE design also expects that the responsibility of representation learning is taken by the encoder through two things: The latent representations of visible patches are not updated in the other parts; and the alignment constraint expects that the predicted representations through latent contextual regressor also lie in the encoded representation space. In comparison to BEiT, MAE, and the approach in the ViT paper, our CAE encoder exploits the greater capability for learning the representation, thus improving the representation quality.

We present the empirical performance of our approach on downstream tasks, semantic segmentation, and object detection and instance segmentation. The results show that our approach outperforms supervised pretraining, contrastive pretraining, and other MIM methods.

## 2 RELATED WORK

Self-supervised representation learning has been widely studied in computer vision , including: context prediction (Doersch et al., 2015; Tian et al., 2021), clustering-based methods (Xie et al., 2016; Yang et al., 2016; Caron et al., 2018; Asano et al., 2019; Zhuang et al., 2019; Huang et al., 2019; Caron et al., 2019; Goyal et al., 2021), contrastive learning (Li et al., 2020; Oord et al., 2018; Henaff, 2020; Wang et al., 2022), instance discrimination (Dosovitskiy et al., 2014; 2015), image discretization (Gidaris et al., 2020a;b), masked image modeling (Li et al., 2021; Fang et al., 2022; Tian et al., 2022), and information maximization (Ermolov et al., 2021; Zbontar et al., 2021; Bardes et al., 2021). The following mainly reviews closely-related methods.

**Autoencoding.** Traditionally, autoencoders were used for dimensionality reduction or feature learning (LeCun, 1987; Gallinari et al., 1987; Hinton & Zemel, 1994; Hinton & Salakhutdinov, 2006; Ranzato et al., 2007; Vincent et al., 2008; Kingma & Welling, 2013). The denoising autoencoder (DAE) is an autoencoder that receives a corrupted data point as input and is trained to estimate the original, uncorrupted data point as its output. The variants or modifications of DAE were adopted for self-supervised representation learning, e.g., corruption by masking pixels (Vincent et al., 2010;

Pathak et al., 2016; Chen et al., 2020a), removing color channels (Zhang et al., 2016), shuffling image patches (Noroozi & Favaro, 2016), denoising pixel-level noise (Atito et al., 2021) and so on.

**Contrastive learning.** In computer vision, contrastive learning[1] has been popular for self-supervised representation learning (Chen et al., 2020b; He et al., 2020; Tian et al., 2020; Chen et al., 2021; Grill et al., 2020; Caron et al., 2021; Chen & He, 2021; Caron et al., 2020; Wu et al., 2018; Peng et al., 2022). The basic idea is to maximize the similarity between the views augmented from the same image and optionally minimize the similarity between the views augmented from different images. Random cropping is an important augmentation scheme, and thus typical contrastive learning methods (e.g., MoCo v3) tend to learn knowledge mainly from the center regions of the original images. Some dense variants (Wang et al., 2021; Xie et al., 2021a) eliminate the tendency in a limited degree by considering an extra contrastive loss with dense patches.

**Masked image modeling.** Motivated by BERT for masked language modeling (Devlin et al., 2019), the method studied in (Dosovitskiy et al., 2021) and BEiT (Bao et al., 2021) use the ViT structure to solve the masked image modeling task, e.g., estimate the pixels or the discrete tokens. But they do not have explicitly an encoder or a decoder and the ViT structure is essentially a mixture of encoder and decoder, limiting the representation learning quality.

Several subsequent MIM methods are developed to improve the encoder quality, such as designing pretraining architectures: iBOT (Zhou et al., 2021), Masked Autoencoder (MAE) (He et al., 2022), SplitMask (El-Nouby et al., 2021), and Simple MIM (SimMIM) (Xie et al., 2021b); adopting new reconstruction targets: Masked Feature Prediction (MaskFeat) (Wei et al., 2021), Perceptual Codebook for BEiT (PeCo) (Dong et al., 2021), and data2vec (Baevski et al., 2022). Our approach belongs to the former one and propose a new pretraining architecture through separating the representation pretraining module and the MIM task completion module and making predictions in the encoded representation space. More about the concurrently-developed methods are provided in Appendix.

## 3 APPROACH

### 3.1 ARCHITECTURE

Our context autoencoder (CAE) pretrains the encoder by solving the masked image modeling task. We randomly split an image into two sets of patches: visible patches $\mathbf{X}_v$ and masked patches $\mathbf{X}_m$. The pretext task is to predict the masked patches from visible patches. The key is to make predictions from visible patches to masked patches in the encoded representation space, and then map the predicted representations of masked patches to the targets. The architecture shown in Figure 1 contains: an encoder, a latent contextual regressor with the alignment constraint, and a decoder.

**Encoder.** The encoder $\mathcal{F}$ maps the visible patches $\mathbf{X}_v$ to the latent representations $\mathbf{Z}_v$. It only handles the visible patches. We use the ViT to form our encoder. It first embeds the visible patches by linear projection as patch embeddings, and adds the positional embeddings $\mathbf{P}_v$. Then it sends the combined embeddings into a sequence of transformer blocks that are based on *self-attention*, generating $\mathbf{Z}_v$.

**Latent contextual regressor.** The latent contextual regressor $\mathcal{H}$ predicts the latent representations $\mathbf{Z}_m$ for the masked patches from the latent representations $\mathbf{Z}_v$ of the visible patches output from the encoder. We form the latent contextual regressor $\mathcal{H}$ using a series of transformer blocks that are based on *cross-attention*.

The initial queries $\mathbf{Q}_m$, called mask queries, are mask tokens that are learned as model parameters and are the same for all the masked patches. The keys and the values are the same and consist of the visible patch representations $\mathbf{Z}_v$ and the output of the previous cross-attention layer (mask queries for the first cross-attention layer). The corresponding positional embeddings are considered when computing the cross-attention weights between the queries and the keys. In this process, the latent representations $\mathbf{Z}_v$ of the visible patches are not updated.

**Alignment constraint.** The latent representation alignment constraint is imposed on the latent representations $\mathbf{Z}_m$ of the masked patches predicted by the latent contextual regressor. We feed the masked patches $\mathbf{X}_m$ into the encoder, which is the same as the one for encoding visible patches, and

---

[1] We use contrastive learning to refer to the self-supervised approach comparing random views with contrastive loss or simply MSE loss that are related as shown in (Garrido et al., 2022).

generate the representations $\bar{\mathbf{Z}}_m$ of the masked patches. We then align the two latent representations $\mathbf{Z}_m$ and $\bar{\mathbf{Z}}_m$ for the masked patches.

**Decoder.** The decoder $\mathcal{G}$ maps the latent representations $\mathbf{Z}_m$ of the masked patches to some forms of the masked patches, $\mathbf{Y}_m$, with the discrete tokens as the targets as done in BEiT. The decoder, similar to the encoder, is a stack of transformer blocks that are based on *self-attention*, followed by a linear layer predicting the targets. The decoder only receives the latent representations of the masked patches (the output of the latent contextual regressor), and the positional embeddings of the masked patches as input without directly using the information of the visible patches.

### 3.2 OBJECTIVE FUNCTION

**Masking and targets.** Following BEiT (Bao et al., 2021), we adopt the random block-wise masking strategy (illustrated in Figure 2) to split the input image into two sets of patches, visible and masked patches. For each image, 98 of 196 ($14 \times 14$) patches are masked.

We form the targets using the discrete tokenizer, e.g., the tokenizer trained with d-VAE on ImageNet-1K without using the labels or the DALL-E tokenizer (trained with d-VAE on 400M images) (Ramesh et al., 2021) used in BEiT Bao et al. (2021). The input image is fed into the tokenizer, assigning a discrete token to each patch. The target tokens for the masked patches are denoted as $\bar{\mathbf{Y}}_m$.

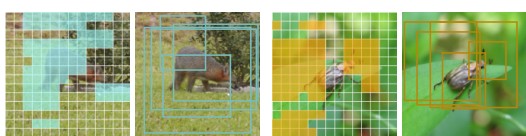

Figure 2: Illustration of random block-wise sampling and random cropping. Random block-wise sampling is used in our approach. Random cropping is a key data-augmentation scheme for contrastive pretraining.

**Loss function.** The loss function consists of a decoding loss: $\ell_y(\mathbf{Y}_m, \bar{\mathbf{Y}}_m)$, and an alignment loss: $\ell_z(\mathbf{Z}_m, \bar{\mathbf{Z}}_m)$. The whole loss is a weighted sum:

$$\ell_y(\mathbf{Y}_m, \bar{\mathbf{Y}}_m) + \lambda \, \ell_z(\mathbf{Z}_m, \text{sg}[\bar{\mathbf{Z}}_m]). \tag{1}$$

We use the MSE loss for $\ell_z(\mathbf{Z}_m, \bar{\mathbf{Z}}_m)$ and the cross-entropy loss for $\ell_y(\mathbf{Y}_m, \bar{\mathbf{Y}}_m)$. $\text{sg}[\cdot]$ stands for stop gradient. $\lambda$ is 2 in our experiments.

## 4 ANALYSIS AND DISCUSSION

**Predictions are made in the encoded representation space.** Our CAE attempts to make predictions in the encoded representation space: predict the representations for the masked patches from the encoded representations of the visible patches. In other words, it is expected that the output representations of the latent contextual regressor also lie in the encoded representation space, which is ensured by an alignment constraint. The constraint about the predicted representation space encourages the learned representation to take on a large extent of semantics for prediction from visible patches to masked patches, benefiting the representation learning of the encoder.

We empirically verify that the predicted representations lie in the encoded representation space through image reconstruction. We train the CAE using the pixel colors as the prediction targets, for two cases: with and without the alignment constraint. For reconstruction, we feed all the patches (without masking, all the image patches are visible) of an image (from the ImageNet validation set) into the pretrained encoder, then skip the latent contextual regressor and directly send all the encoded patch representations to the pretrained decoder for reconstructing the whole image.

Figure 3 provides reconstruction results for several examples randomly sampled from the ImageNet-1K validation set. One can see that our approach can successfully reconstruct the images, implying that the input and output representations of latent contextual regressor are in the same space. In contrast, without the alignment constraint, the reconstructed images are noisy, indicating the input and output representations of latent contextual regressor are in the different spaces. The results suggest that the alignment constraint is critical for ensuring that predictions are made in the encoded representation space.

**Representation alignment in CAE and contrastive learning.** Representation alignment is also used in contrastive learning methods, such as MoCo, BYOL, SimCLR, and methods mixing contrastive

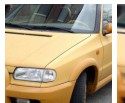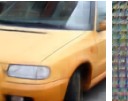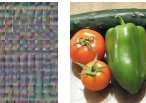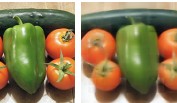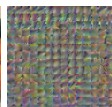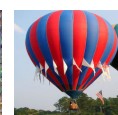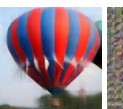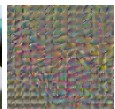

Figure 3: Illustrating that predictions are made in the representation space. We reconstruct the image by feeding the full image (1st, 4th, and 7th) into the pretrained CAE encoder and then the pretrained CAE decoder outputting the reconstructed image (2nd, 5th, and 8th). It can be seen that the image can be constructed with the semantics kept when skipping latent contextual regressor, verifying the input and the predicted representations lie in the same space. We also show the reconstructed images (3rd, 6th, and 9th) from the encoder and the decoder pretrained without the alignment constraint. We can see that those images are meaningless, indicating that the alignment constraint is critical for ensuring that predictions are made in the representation space.

learning and masked image modeling, such as iBOT, and MST, and the alignment loss could be the MSE loss or the contrastive loss that CAE may aslo take advantage of. In the CAE, the alignment is imposed over the representations $\mathbf{Z}_m = \mathcal{H}(\mathcal{F}(\mathbf{X}_v)$ predicted from the representations $\mathcal{F}(\mathbf{X}_v)$ of visible patches through the regressor $\mathcal{H}$, and the representations $\bar{\mathbf{Z}}_m = \mathcal{F}(\mathbf{X}_m)$ computed from the encoder $\mathcal{F}$, *both about the masked patches*. Differently, the alignment in contrastive learning is imposed over the representations $\{\mathcal{F}(\mathbf{V}_1), \mathcal{F}(\mathbf{V}_2), \cdots, \mathcal{F}(\mathbf{V}_N)\}$ of *different views* $\{\mathbf{V}_1, \mathbf{V}_2, \cdots, \mathbf{V}_N\}$.

**Relation to autoencoder.** The original autoencoder (LeCun, 1987; Gallinari et al., 1987; Hinton & Zemel, 1994) consists of an encoder and a decoder. The encoder maps the input into a latent representation, and the decoder reconstructs the input from the latent representation. The denoising autoencoder (DAE) (Vincent et al., 2010), a variant of autoencoder, corrupts the input by adding noises and still reconstructs the non-corrupted input.

Our CAE encoder is similar to the original autoencoder and also contains an encoder and a decoder. Different from the autoencoder where the encoder and the decoder process the whole image, our encoder takes a portion of patches as input and our decoder takes the estimated latent representations of the other portion of patches as input. Importantly, the CAE introduces a latent contextual regressor that makes predictions in the latent space from the visible patches to the masked patches.

**Relation to BEiT and MAE.** The CAE encoder processes the visible patches, to extract their representations, without making predictions for masked patches. Latent contextual regressor does not update the representations for visible patches: the representations of the visible patches in the regressor are the values and keys for cross-attention; the alignment constraint expects that the output of latent contextual regressor lies in the representation space same with the encoder output. The decoder only processes the predicted representations of masked patches. Therefore, the encoder takes the responsibility of and is only for representation learning.

In contrast, BEiT (Bao et al., 2021) and the MIM part of iBOT do not separate the representation extraction role and the task completion role and uses a single network, with both the visible and masked patches as the input, simultaneously for the two roles. In MAE (He et al., 2022), the so-called decoder may play a partial role for representation learning as the representations of the visible patches are also updated in the MAE decoder. Unlike CAE, MAE does not explicitly regress the representations (that lie in the space the same as the encoded representation space) for masked patches.

When the pretrained encoder is applied to downstream tasks, one often replaces the pretext task completion part using the downstream task layer, e.g., segmentation layer or detection layer. The separation of representation learning (encoding) and pretext task completion helps that downstream task applications take good advantage of representation pretraining.

**Comparison to contrastive learning.** Typical contrastive learning methods, e.g., SimCLR (Chen et al., 2020b) and MoCo (He et al., 2020; Chen et al., 2021), pretrain the networks by solving the pretext task, maximizing the similarities between augmented views (e.g., random crops) from the same image and minimizing the similarities between augmented views from different images.

It is shown in (Chen et al., 2020b) that random cropping plays an important role in view augmentation for contrastive learning. Through analyzing random crops (illustrated in Figure 2), we observe that the center pixels in the original image space have large chances to belong to random crops. We suspect that the global representation, learned by contrastive learning for a random crop possibly with other augmentation schemes, tends to focus mainly on the center pixels in the original image, so

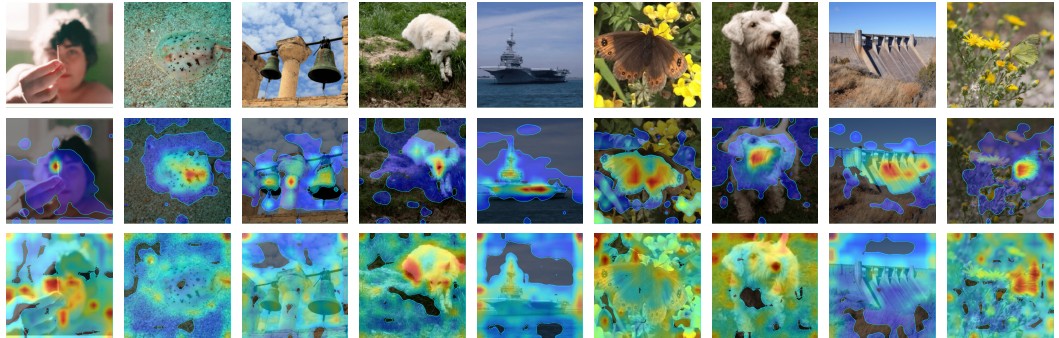

Figure 4: Illustrating the attention map averaged over 12 attention heads between the class token and the patch tokens in the last layer of the ViT encoder pretrained on ImageNet-1K. The region inside the blue contour is obtained by thresholding the attention weights to keep 50% of the mass. Top: Input image, Middle: MoCo v3, a typical contrastive learning method, and Bottom: our CAE. One can see that MoCo v3 tends to focus mainly on the centering regions and little on other patches, and our CAE tends to consider almost all the patches.

that the representations of different crops from the same image can be possibly similar. Figure 4 (the second row) shows that the center region of the original image for the typical contrastive learning approach, MoCo v3, is highly attended.

In contrast, our CAE method (and other MIM methods) randomly samples the patches from the augmented views to form the visible and masked patches. All the patches are possible to be randomly masked for the augmented views and accordingly the original image. Thus, the CAE encoder needs to learn good representations for all the patches, to make good predictions for the masked patches from the visible patches. Figure 4 (the third row) illustrates that almost all the patches in the original images are considered in our CAE encoder.

Considering that the instances of the 1000 categories in ImageNet-1K locate mainly around the center of the original images (Russakovsky et al., 2015), typical contrastive learning methods, e.g., MoCo v3, learn the knowledge mainly about the 1000 categories, which is similar to supervised pretraining. But our CAE and other MIM methods are able to learn more knowledge beyond the 1000 categories from the non-center image regions. This indicates that the CAE has the potential to perform better for downstream tasks.

## 5 EXPERIMENTS

### 5.1 IMPLEMENTATION

We study the standard ViT small, base and large architectures, ViT-S (12 transformer blocks with dimension 384), ViT-B (12 transformer blocks with dimension 768) and ViT-L (24 transformer blocks with dimension 1024). The latent contextual regressor consists of 4 transformer blocks based on cross-attention, and the decoder consists of 4 transformer blocks based on self-attention, and an extra linear projection for making predictions.

We train the CAE on ImageNet-1K. We partition the image of $224 \times 224$ into $14 \times 14$ patches with the patch size being $16 \times 16$. We use standard random cropping and horizontal flipping for data augmentation. The pretraining settings are almost the same as BEiT (Bao et al., 2021). (See Appendix for details).

### 5.2 PRETRAINING EVALUATION

**Linear probing.** Linear probing is widely used as a proxy of pretraining quality evaluation for self-supervised representation learning. It learns a linear classifier over the image-level representation output from the pretrained encoder by using the labels of the images, and then tests the performance on the validation set.

**Attentive probing.** The output of the encoder pretrained with MIM methods are representations for all the patches. It is not suitable to linearly probe the representation, averagely-pooled from patch

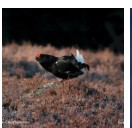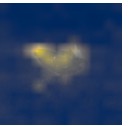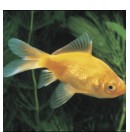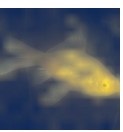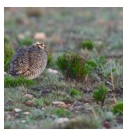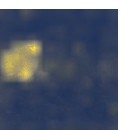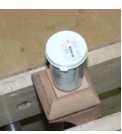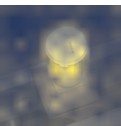

Figure 5: Illustrating the cross-attention unit in attentive probing. The attention map is the average of cross-attention maps over 12 heads between the extra class token and the patches. One can see that the attended region lies mainly in the object, which helps image classification.

representations, because the image label in ImageNet-1K only corresponds to a portion of patches. It is also not suitable to use the default class token within the encoder because the default class token serves as a role of aggregating the patch representations for better patch representation extraction and is not merely for the portion of patches corresponding to the image label.

To use the image-level label as a proxy of evaluating the pretraining quality for the encoder pretrained with MIM methods, we need to attend the patches that are related to the label. We introduce a simple modification by using a cross-attention unit with an extra class token (that is different from the class token in the encoder) as the query and the output patch representations of the encoder as the keys and the values, followed by a linear classifier. The introduced cross-attention unit is able to care mainly about the patches belonging to the 1000 classes in ImageNet-1K and remove the interference of other patches. Figure 5 illustrates the effect of the cross-attention unit, showing that the extra cross-attention unit can to some degree attend the regions that are related to the 1000 ImageNet-1K classes.

**Results.** Table 1 shows the results with three schemes, linear probing (LIN), attentive probing (ATT), and fine-tuning (FT) for representative contrastive pretraining (MoCo v3 and DINO) and MIM (BEiT and MAE) methods, as well as our approach with the targets formed with the DALL-E tokenizer (trained on 400M images) and the d-VAE tokenizer (trained on ImageNet-1K without using the labels), denoted as CAE* and CAE, respectively. The models of MAE with 300 epochs and BEiT are pretrained by us using the official implementations, and other models are the officially released models.

We highlight a few observations. The fine-tuning performance for these methods are very similar and there is only a minor difference similar to the observation (Zhou et al., 2021). We think that the reason is that self-supervised pretraining and fine-tuning are conducted on the same dataset and no extra knowledge is introduced for image classification. The minor difference might come from the optimization aspect: different initialization (provided by pretrained models) for fine-tuning.
In terms of linear probing, the scores of the contrastive learning methods, MoCo v3 and DINO, are higher than the MIM methods. This is as expected because contrastive learning focuses mainly on learning the representations for 1000 classes (See discussion in Section 4). The pretraining is relatively easier than existing MIM methods as contrastive learning mainly cares about the 1000 classes and MIM methods may care about the classes beyond the 1000 classes.

For the MIM methods, the scores of attentive probing are much larger than linear probing. This validates our analysis: the MIM methods extract the representations for all the patches, and the classification task needs to attend the corresponding portion of patches.

The LIN and ATT scores are similar for contrastive pretraining: e.g., with ViT-B, (76.2 vs 77.0) for MoCo v3, and (77.3 vs 77.8) for DINO. This means that the extra cross-attention in attentive probing does not make a big difference, which is one more evidence for our analysis in Section 4 that they already focus mainly on the region where the instance in the 1000 categories lies.

## 5.3 ABLATION STUDIES

The CAE architecture contains three components for pretraining the encoder: latent contextual regressor, decoder, and alignment constraint. We cannot remove the latent contextual regressor that is the only unit to make predictions for masked patches from visible patches in our architecture. We study the other two components, the decoder (when the decoder is removed, we use a linear layer to predict the targets) and the alignment constraint.

Table 2 shows the ablation results. We report the scores for attentive probing, and downstream tasks: semantic segmentation on ADE20K and object detection on COCO with the DALL-E tokenizer as the

Table 1: Pretraining quality evaluation in terms of fine-tuning (FT), linear probing (LIN), and attentive probing (ATT). ‡ means the number of effective epochs in (Zhou et al., 2021) as they adopt multi-crop augmentation (equivalently take a larger number of epochs compared to one-crop augmentation). We report the top-1 accuracy (in the column ATT) of the supervised training approach DeiT (Touvron et al., 2020) to show how far the ATT score is from supervised training. The scores for other models and our models are based on our implementations if not specified. Except that * denotes using the DALL-E tokenizer, CAE adopts the d-VAE tokenizer trained on ImageNet-1K only. †: these results are from (He et al., 2022).

| Method | #Epochs | #Crops | FT | LIN | ATT |
|---|---|---|---|---|---|
| *Methods using ViT-S:* | | | | | |
| DeiT | 300 | - | - | - | 79.9 |
| MoCo v3 | 600‡ | 2 | 81.7 | 73.1 | 73.8 |
| BEiT | 300 | 1 | 81.7 | 15.7 | 23.6 |
| CAE* | 300 | 1 | **82.0** | 51.8 | 65.0 |
| *Methods using ViT-B:* | | | | | |
| DeiT | 300 | - | - | - | 81.8 |
| MoCo v3 | 600‡ | 2 | 83.0 | 76.2 | 77.0 |
| DINO | 1600‡ | 12 | 83.3 | 77.3 | 77.8 |
| BEiT | 300 | 1 | 83.0 | 37.6 | 49.4 |
| MAE | 300 | 1 | 82.9 | 61.5 | 71.1 |
| MAE | 1600 | 1 | 83.6 | 67.8 | 74.2 |
| iBOT | 1600‡ | 12 | 83.8 | 79.5 | 79.8 |
| CAE* | 300 | 1 | 83.6 | 64.1 | 73.8 |
| CAE* | 800 | 1 | 83.8 | 68.6 | 75.9 |
| CAE* | 1600 | 1 | **83.9** | 70.4 | 77.1 |
| CAE | 1600 | 1 | **83.9** | 71.4 | 77.4 |
| *Methods using ViT-L:* | | | | | |
| MoCo v3† | 600‡ | 2 | 84.1 | - | - |
| BEiT† | 1600 | 1 | 85.2 | - | - |
| MAE | 1600 | 1 | 86.0 | 76.0 | 78.8 |
| CAE* | 1600 | 1 | **86.3** | 78.1 | 81.2 |
| CAE | 1600 | 1 | **86.3** | 77.9 | 81.2 |

Table 2: Ablation studies for the decoder and the alignment constraint in our CAE. All the models are pretrained on ImageNet-1K with 300 epochs.

| | Decoder | Align | ATT | ADE | COCO |
|---|---|---|---|---|---|
| CAE* | × | × | 71.2 | 47.0 | 46.9 |
| CAE* | √ | × | 72.7 | 47.1 | 47.2 |
| CAE* | √ | √ | 73.8 | 48.3 | 48.4 |

Table 3: Semantic segmentation on ADE20K. All the results are based on the same implementation for semantic segmentation. #Epochs refers to the number of pretraining epochs. ‡ means the number of effective epochs in (Zhou et al., 2021) as the method uses multi-crop pretraining augmentation (See Table 1). †: these results are from (He et al., 2022).

| Method | #Epochs | mIoU |
|---|---|---|
| *Methods using ViT-B:* | | |
| DeiT | 300 | 47.0 |
| MoCo v3 | 600‡ | 47.2 |
| DINO | 1600‡ | 47.2 |
| BEiT | 300 | 45.5 |
| BEiT | 800 | 46.5 |
| MAE | 300 | 45.8 |
| MAE | 1600 | 48.1 |
| iBOT | 1600‡ | 50.0 |
| CAE* | 300 | 48.3 |
| CAE* | 800 | 49.7 |
| CAE* | 1600 | **50.2** |
| CAE | 1600 | 50.1 |
| *Methods using ViT-L:* | | |
| MoCo v3† | 600‡ | 49.1 |
| BEiT† | 1600 | 53.3 |
| MAE | 1600 | 53.6 |
| CAE* | 1600 | **54.7** |
| CAE | 1600 | 54.6 |

targets. One can see that the downstream task performance is almost the same when only the decoder is added and that the performance increases when the decoder and the alignment constraint are both added. This also verifies that the alignment constraint is important for ensuring that the predicted representations of masked patches lie in the encoded representation space and thus the predictions are made in the encoded representation space, and accordingly improving the representation quality.

## 5.4 DOWNSTREAM TASKS

**Semantic segmentation on ADE20K** (Zhou et al., 2017). We follow the implementation (Bao et al., 2021) to use UperNet (Xiao et al., 2018) (Appendix for training details). In both downstream tasks, the CAE with the tokenizers learned over ImageNet-1K performs almost the same as the tokenizers learned over 400M images provided by DALL-E (CAE*), implying that the tokenizer trained on ImageNet-1K (without using the labels) or a larger dataset does not affect the pretraining quality and accordingly the downstream task performance.

Table 4: Object detection and instance segmentation on COCO. Mask R-CNN is adopted and trained with the $1\times$ schedule. All the results are based on the same implementation for object detection and instance segmentation. #Epochs refers to the number of pretraining epochs on ImageNet-1K. $\ddagger$ means the number of effective epochs in (Zhou et al., 2021) (See Table 1).

| Method | #Epochs | Supervised | Self-supervised | Object detection | | | Instance segmentation | | |
|---|---|---|---|---|---|---|---|---|---|
| | | | | $AP^b$ | $AP^b_{50}$ | $AP^b_{75}$ | $AP^m$ | $AP^m_{50}$ | $AP^m_{75}$ |
| *Methods using ViT-S:* | | | | | | | | | |
| DeiT | 300 | $\checkmark$ | $\times$ | 43.1 | 65.2 | 46.6 | 38.4 | 61.8 | 40.6 |
| MoCo v3 | 600$^\ddagger$ | $\times$ | $\checkmark$ | 43.3 | 64.9 | 46.8 | 38.8 | 61.6 | 41.1 |
| BEiT | 300 | $\times$ | $\checkmark$ | 35.6 | 56.7 | 38.3 | 32.6 | 53.3 | 34.2 |
| CAE* | 300 | $\times$ | $\checkmark$ | **44.1** | 64.6 | 48.2 | **39.2** | 61.4 | 42.2 |
| *Methods using ViT-B:* | | | | | | | | | |
| DeiT | 300 | $\checkmark$ | $\times$ | 46.9 | 68.9 | 51.0 | 41.5 | 65.5 | 44.4 |
| MoCo v3 | 600$^\ddagger$ | $\times$ | $\checkmark$ | 45.5 | 67.1 | 49.4 | 40.5 | 63.7 | 43.4 |
| DINO | 1600$^\ddagger$ | $\times$ | $\checkmark$ | 46.8 | 68.6 | 50.9 | 41.5 | 65.3 | 44.5 |
| BEiT | 300 | $\times$ | $\checkmark$ | 39.5 | 60.6 | 43.0 | 35.9 | 57.7 | 38.5 |
| BEiT | 800 | $\times$ | $\checkmark$ | 42.1 | 63.3 | 46.0 | 37.8 | 60.1 | 40.6 |
| MAE | 300 | $\times$ | $\checkmark$ | 45.4 | 66.4 | 49.6 | 40.6 | 63.4 | 43.7 |
| MAE | 1600 | $\times$ | $\checkmark$ | 48.4 | 69.4 | 53.1 | 42.6 | 66.1 | 45.9 |
| iBOT | 1600$^\ddagger$ | $\times$ | $\checkmark$ | 48.2 | 69.7 | 52.8 | 42.7 | 66.5 | 46.0 |
| CAE* | 300 | $\times$ | $\checkmark$ | 48.4 | 69.2 | 52.9 | 42.6 | 66.1 | 45.8 |
| CAE* | 800 | $\times$ | $\checkmark$ | 49.8 | 70.7 | 54.6 | 43.9 | 67.8 | 47.4 |
| CAE* | 1600 | $\times$ | $\checkmark$ | 50.0 | 70.9 | 54.8 | 44.0 | 67.9 | 47.6 |
| CAE | 1600 | $\times$ | $\checkmark$ | **50.2** | 71.0 | 54.9 | **44.2** | 68.3 | 47.9 |
| *Methods using ViT-L:* | | | | | | | | | |
| MAE | 1600 | $\times$ | $\checkmark$ | 54.0 | 74.3 | 59.5 | 47.1 | 71.5 | 51.0 |
| CAE* | 1600 | $\times$ | $\checkmark$ | 54.5 | 75.2 | 60.1 | 47.6 | 72.2 | 51.9 |
| CAE | 1600 | $\times$ | $\checkmark$ | **54.6** | 75.2 | 59.9 | **47.6** | 72.0 | 51.9 |

Table 3 shows that using the ViT-B, our CAE* with 300 training epochs performs better than DeiT, MoCo v3, DINO, MAE (1600 epochs) and BEiT. Our CAE* (1600 epochs) further improves the segmentation scores and outperforms MAE (1600 epochs), MoCo v3 and DeiT by 2.1, 3.0 and 3.2, respectively. Using ViT-L, our CAE* (1600 epochs) outperforms BEiT (1600 epochs) and MAE (1600 epochs) by 1.4 and 1.1, respectively.

The superior results over supervised and contrastive pretraining methods, DeiT, MoCo v3 and DINO, stem from that our approach captures the knowledge beyond the 1000 classes in ImageNet-1K. The superior results over BEiT and MAE stem from that our CAE makes predictions in the encoded representation space and that representation learning and pretext task completion are separated.

**Object detection and instance segmentation on COCO** (Lin et al., 2014). We adopt the Mask R-CNN approach (He et al., 2017) that produces bounding boxes and instance masks simultaneously, with the ViT as the backbone (see Appendix for training details). The results are given in Table 4. We report the box AP for object detection and the mask AP for instance segmentation. The observations are consistent with those for semantic segmentation in Table 3. Our CAE* (300 epochs, ViT-B) is superior to all the other models except that a little lower than MAE (1600 epochs). Our approach (1600 epochs) outperforms MAE (1600 epochs), MoCo v3 and DeiT by 1.6, 4.5 and 3.1, respectively. Using ViT-L, our CAE achieves 54.6 box AP and outperforms MAE by 0.6.

## 6 CONCLUSION

The core design of our CAE architecture for masked image modeling is that predictions are made from visible patches to masked patches in the encoded representation space. Experiments demonstrate the effectiveness of our design. In addition, we also point out that the advantage of MIM methods over typical contrastive pretraining and supervised pretraining on ImageNet-1K is that MIM learns the representations for all the patches, while typical contrastive pretraining (e.g., MoCo and SimCLR) and supervised pretraining tend to learn semantics mainly from center patches of the original images and little from non-center patches.

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

# APPENDIX

## A   TRAINING DETAILS

**Pretraining.** The settings are almost the same as BEiT (Bao et al., 2021). We use AdamW (Loshchilov & Hutter, 2017) for optimization and train the CAE for 300/800/1600 epochs with the batch size being 2048. We set the learning rate as 1.5e-3 with cosine learning rate decay. The weight decay is set as 0.05. The warmup epochs for 300/800/1600 epochs pre-training are 10/20/40, respectively. We employ drop path (Huang et al., 2016) rate 0.1 and dropout rate 0.

**Fine-tuning on ImageNet.** We follow the fine-tuning protocol in BEiT to use layer-wise learning rate decay, weight decay and AdamW. The batch size is 4096, the warmup epoch is 5 and the weight decay is 0.05. For ViT-S, we train 200 epochs with learning rate 1.6e-2 and layer-wise decay rate 0.75. For ViT-B, we train 100 epochs with learning rate 8e-3 and layer-wise decay rate 0.65. For ViT-L, we train 50 epochs with learning rate 2e-3 and layer-wise decay rate 0.75.

**Linear probing.** We use the LARS (You et al., 2017) optimizer with momentum 0.9. The model is trained for 90 epochs. The batch size is 16384, the warmup epoch is 10 and the learning rate is 6.4. Following (He et al., 2022), we adopt an extra BatchNorm layer (Ioffe & Szegedy, 2015) without affine transformation (`affine=False`) before the linear classifier. We do not use mixup (Zhang et al., 2017), cutmix (Yun et al., 2019), drop path (Huang et al., 2016), or color jittering, and we set weight decay as zero.

**Attentive probing.** The parameters of the encoder are fixed during attentive probing. A cross-attention module, a BatchNorm layer (`affine=False`), and a linear classifier are appended after the encoder. The extra class token representation in cross-attention is learned as model parameters. The keys and the values are the patch representations output from the encoder. There is no MLP or skip connection operation in the extra cross-attention module. We use the SGD optimizer with momentum 0.9 and train the model for 90 epochs. The batch size is 8192, the warmup epoch is 10 and the learning rate is 0.4. Same as linear probing, we do not use mixup (Zhang et al., 2017), cutmix (Yun et al., 2019), drop path, or color jittering, and we set weight decay as zero.

**Hyperparameter choice.** There is a tradeoff variable $\lambda$ in the loss function given in Equation 1 in the main paper. We did not do an extensive study and only tried three choices, $\lambda = 1$, $\lambda = 1.5$ and $\lambda = 2$. The linear probing results are 63.4, 63.7 and 64.1, respectively. The choice $\lambda = 1$ works also well, slightly worse than $\lambda = 2$ that is adopted in our experiment.

We also conduct experiments with different mask ratios including 40%, 50%, and 60%. Results are listed in Table 5. We find that ratio 50% gets better results than ratio 40%. Adopting a higher mask ratio (60%) could further improve the performance of linear probing and attentive probing, while the semantic segmentation performance is reduced by 0.2%. We choose 50% in our work unless specified.

Table 5: Ablation study on different mask ratios. ViT-B is used here.

| Mask Ratio | LIN | ATT | ADE Seg |
|---|---|---|---|
| 40% | 63.1 | 73.0 | 47.2 |
| 50% | 64.1 | 73.8 | 48.3 |
| 60% | 64.8 | 74.2 | 48.1 |

Table 6: Ablation study on number of layers in the latent contextual regressor and decoder. 4 layers means we use 4 layers in latent contextual regressor and 4 layers in decoder. ViT-B is used here.

| Layer Num | LIN | ATT |
|---|---|---|
| 1 | 58.7 | 67.5 |
| 2 | 62.1 | 71.7 |
| 4 | 64.1 | 73.8 |
| 5 | 64.2 | 73.7 |

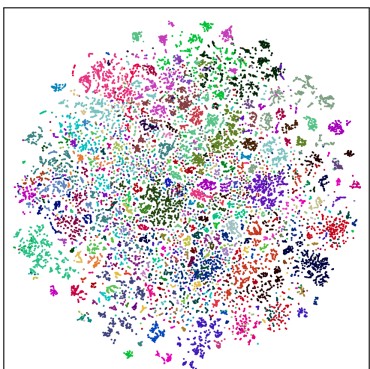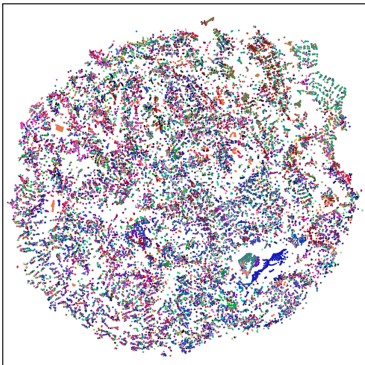

Figure 6: t-SNE visualization (one color for one category) of representations extracted from the images in ADE20K. Left: ViT pretrained with our CAE; Right: ViT with random weights.

For the number of layers in the latent contextual regressor and decoder, we tried four choices: 1-layer, 2-layers, 4-layer and 5-layer. Results are listed in Table 6. We empirically observed that 4-layer outperforms other choices overall.

**Object detection and instance segmentation on COCO.** We utilize multi-scale training and resize the image with the size of the short side between $480$ and $800$ and the longe side no larger than $1333$. The batch size is $32$. For the ViT-S, the learning rate is 3e-4, the layer-wise decay rate is $0.75$, and the drop path rate is $0.1$. For the ViT-B, the learning rate is 3e-4, the layer-wise decay rate is $0.75$, and the drop path rate is $0.2$. For the ViT-L, the learning rate is 2e-4, the layer-wise decay rate is $0.8$, and the drop path rate is $0.2$. We train the network with the $1\times$ schedule: $12$ epochs with the learning rate decayed by $10\times$ at epochs $9$ and $11$. We do not use multi-scale testing. The Mask R-CNN implementation follows MMDetection (Chen et al., 2019).

**Semantic segmentation on ADE20K.** We use AdamW as the optimizer. The input resolution is $512 \times 512$. The batch size is $16$. For the ViT-B, the layer-wise decay rate is $0.65$ and the drop path rate is $0.1$. We search from four learning rates, 1e-4, 2e-4, 3e-4 and 4e-4, for all the results in Table 3 in the main paper. For the ViT-L, the layer-wise decay rate is $0.95$ and drop path rate is $0.15$. We search from three learning rates for all the methods, 3e-5, 4e-5 and 5e-5, We conduct fine-tuning for $160$K steps. We do not use multi-scale testing.

## B    INTERPRETATION

**Intuitive Interpretation for CAE.** Humans are able to hallucinate what appears in the masked regions and how they appear according to the visible regions. We speculate that humans do this possibly in a way similar as the following example: given that only the region of the dog's head is visible and the remaining parts are missing, one can (a) recognize the visible region to be about a dog, (b) predict the regions where the other parts of the dog appear, and (c) guess what the other parts look like.

Our CAE encoder is in some sense like the human recognition step (a). It understands the content by mapping the visual patches into latent representations that lie in the subspace that corresponds to the category dog[2]. The latent contextual regressor is like step (b). It produces a plausible hypothesis for the masked patches, and describes the regions corresponding to the other parts of the dog using latent representations. The CAE decoder is like step (c), mapping the latent representations to the targets. It should be noted that the latent representations might contain other information besides the semantic information, e.g., the part information and the information for making predictions.

We adopt t-SNE (Van der Maaten & Hinton, 2008) to visualize the high-dimensional patch representations output from our CAE encoder on ADE20K (Zhou et al., 2017) in Figure 6. ADE20K has a total of $150$ categories. For each patch in the image, we set its label to be the category that more than half of the pixels belong to. We collect up to $1000$ patches for each category from sampled $500$ images.

---

[2]Our encoder does not know that the subspace is about a dog, and just separates it from the subspaces of other categories.

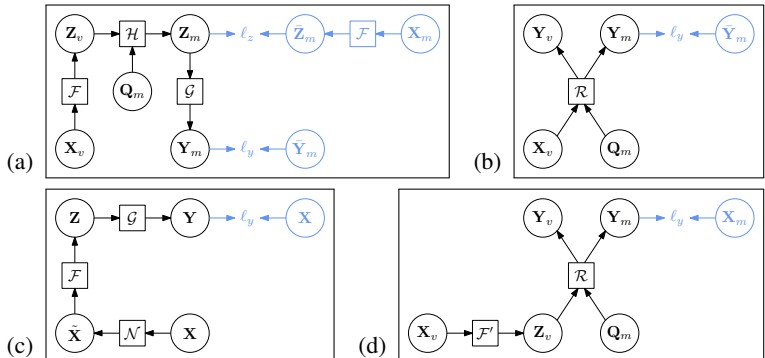

Figure 7: The computational graphs for (a) a context autoencoder (CAE), (b) BEiT (Bao et al., 2021), (c) a denoising autoencoder (DAE), and (d) MAE (He et al., 2022) and the one stream in SplitMask (El-Nouby et al., 2021). The parts in cornflower blue are for loss function. (a) The encoder $\mathcal{F}$ receives visible patches $\mathbf{X}_v$ and outputs their latent representations $\mathbf{Z}_v$. The latent contextual regressor $\mathcal{H}$ predicts the latent representations $\mathbf{Z}_m$ for masked patches from $\mathbf{Z}_v$. The decoder predicts the targets $\mathbf{Y}_m$ for masked patches from $\mathbf{Z}_m$. $\ell_z$ and $\ell_y$ are the loss functions. During training, the gradient is stopped for $\bar{\mathbf{Z}}_m$. See the detail in Section 3 in the main paper. (b) The input includes both visible patches $\mathbf{X}_v$ and mask queries $\mathbf{Q}_m$ representing masked patches, and the representations for them are updated within the function $\mathcal{R}$. (c) The function $\mathcal{N}$ is a noising function generating the noisy version $\tilde{\mathbf{X}}$ from the input $\mathbf{X}$. $\mathcal{F}$ and $\mathcal{G}$ are the normal encoder and decoder, respectively. (d) The two functions, $\mathcal{F}'$ and $\mathcal{R}$, are both based on self-attention. $\mathcal{F}'$ (called encoder in MAE) only processes the visible patches $\mathbf{X}_v$, and $\mathcal{R}$ (called decoder in MAE) processes both the latent representations $\mathbf{Z}_v$ of the visible patches and the mask queries ($\mathbf{Q}_m$) and updates them simultaneously. For simplicity, the positional embeddings are not included in computational graphs. *(a) CAE and (c) DAE perform the encoding and MIM task completion roles explicitly and separately, (b) BEiT and (d) MAE perform the encoding and MIM task completion roles implicitly and simultaneously.*

As shown in the figure, the latent representations of CAE are clustered to some degree for different categories (though not perfect as our CAE is pretrained on ImageNet-1K). Similar observations could be found for other MIM methods.

**Probabilistic interpretation for CAE.** The MIM problem can be formulated in the probabilistic form, maximizing the probability of the predictions $\mathbf{Y}_m$ of the masked patches given the conditions, the visible patches $\mathbf{X}_v$, the positions $\mathbf{P}_v$ of the visible patches, and the positions $\mathbf{P}_m$ of the masked patches: $P(\mathbf{Y}_m|\mathbf{X}_v, \mathbf{P}_v, \mathbf{P}_m)$. It can be solved by introducing latent representations $\mathbf{Z}_m$ and $\mathbf{Z}_v$, with the assumption that $\mathbf{Z}_v$ and $\mathbf{P}_m$ ($\mathbf{Y}_m$ and $\mathbf{P}_v$) are conditionally independent:

$$P(\mathbf{Y}_m|\mathbf{X}_v, \mathbf{P}_v, \mathbf{P}_m) = P(\mathbf{Z}_v|\mathbf{X}_v, \mathbf{P}_v)P(\mathbf{Z}_m|\mathbf{Z}_v, \mathbf{P}_v, \mathbf{P}_m)P(\mathbf{Y}_m|\mathbf{Z}_m, \mathbf{P}_m),$$

where the three terms on the right side correspond to three parts of our CAE: the encoder, the latent contextual regressor, and the decoder, respectively.

The latent representation alignment constraint can be written as a conditional probability, $P(\mathbf{Z}_m|\bar{\mathbf{Z}}_m)$, where $\bar{\mathbf{Z}}_m$ is the masked patch representations computed from the encoder.

**Intuitive interpretation for contrastive learning.** We consider the case in ImageNet-1K that the object mainly lies in the center of an image[3]. There are $N$ randomly sampled crops from an image, and each crop $\mathbf{I}_n$ contains a part of the center object, $\mathbf{O}_n$. To maximize the similarity between two crops $\mathbf{I}_m$ and $\mathbf{I}_n$, the pretraining might contain the processes: Select the regions $\mathbf{O}_m$ and $\mathbf{O}_n$ from the two crops $\mathbf{I}_m$ and $\mathbf{I}_n$, extract their features $\mathbf{f}_{om}$ and $\mathbf{f}_{on}$, and predict the feature of the object, $\mathbf{f}_o$, from the part features $\mathbf{f}_{om}$ and $\mathbf{f}_{on}$. In this way, the features of the crops from the same image could be similar. Among the $N$ random crops, most crops contain a part of the object in the center, and a few crops that do not contain a part of the center object could be viewed as noises when optimizing the contrastive loss.

---

[3]There are a few images in which the object does not lie in the center in ImageNet-1K. The images are actually viewed as noises and have little influence for contrastive learning.

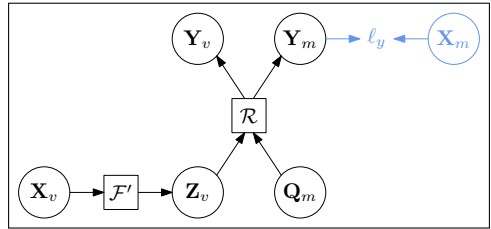

Figure 8: The computational graph for MAE (He et al., 2022) and the one stream in SplitMask (El-Nouby et al., 2021). The two functions, $\mathcal{F}'$ and $\mathcal{R}$, are both based on self-attention. $\mathcal{F}'$ (called encoder in MAE) only processes the visible patches $\mathbf{X}_v$, and $\mathcal{R}$ (called decoder in MAE) processes both the latent representations $\mathbf{Z}_v$ of the visible patches and the mask queries ($\mathbf{Q}_m$) and updates them simultaneously.

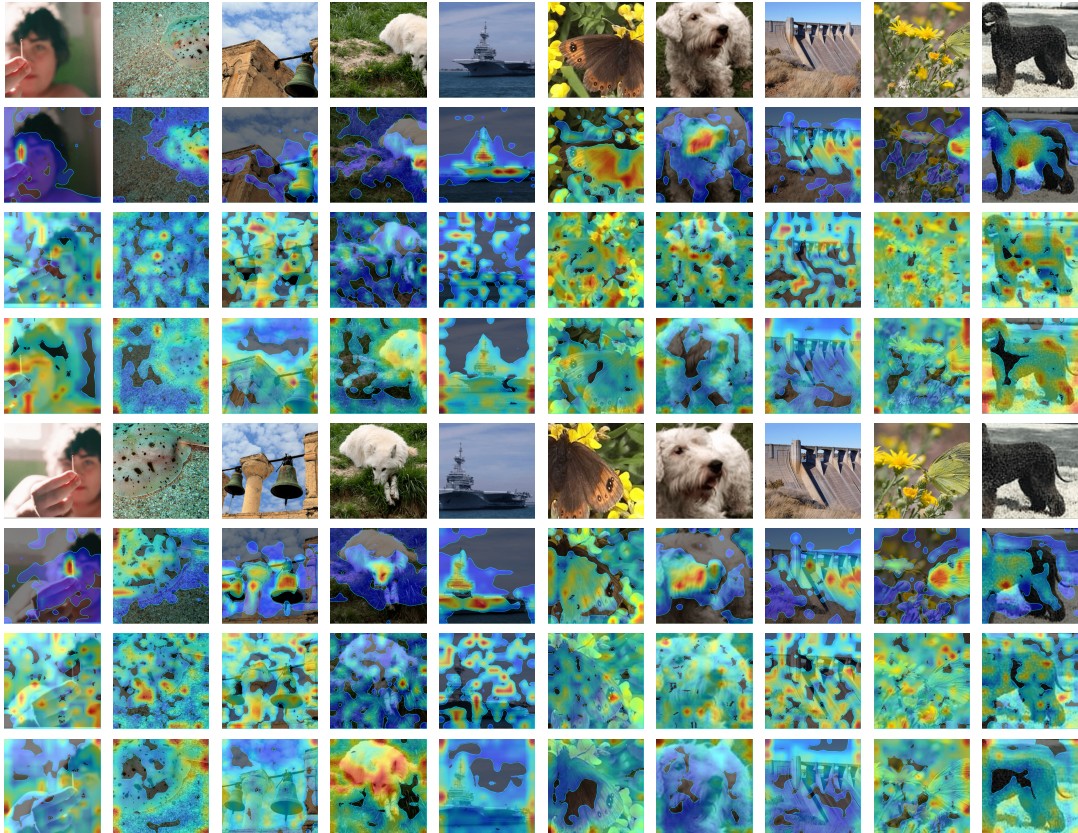

Figure 9: The attention maps over two sets of randomly cropped images (the 1st the 5th rows) for MoCo v3 (the 2nd the 6th rows), MAE (the 3rd the 7th rows), and our CAE (the 4th the 8th rows) pretrained on ImageNet-1K. The contrastive learning method, MoCo v3, tends to focus mainly on the object region and little on other regions. However, MIM-based models, CAE and MAE, tend to consider almost all the patches. The attention maps over the original images are shown in Figure 4 in the main paper.

After pretrained on ImageNet-1K (where the object mainly lies in the center) the encoder is able to learn the knowledge of the 1000 classes and localize the region containing the object belonging to the 1000 classes. It is not necessary that the object lies in the center for the testing image. We show the attention maps of MoCo v3 and our CAE for random crops in Figure 9. This further verifies that MoCo v3 (contrastive pretraining) pretrained on ImageNet-1K tends to attend to the object region, corresponding to the center region of the original image as shown in Figure 4 in the main paper.

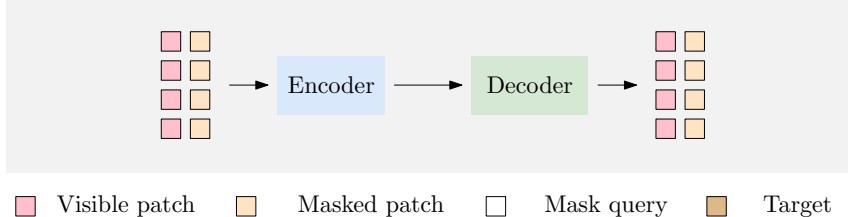

Figure 10: The architecture for image reconstruction by skipping the latent regressor. After we pretrain the CAE model, we feed the full image (including visible patches and masked patches) to the pretrained CAE encoder and then the decoder by skipping the latent regressor.

**Computational graph.** We provide the computational graph for CAE, BEiT (Bao et al., 2021), denoising autoencoder, Masked autoencoder (He et al., 2022) and SplitMask (El-Nouby et al., 2021) (one stream) in Figure 7. Compared to our CAE, the main issue of MAE is that the so-called decoder $\mathcal{R}$ might have also the encoding role, i.e., learning semantic representations of the visible patches.

**Image reconstruction by skipping the latent regressor.** To verify that the input and output representation of latent contextual regressor lie in the same space, we conduct experiments on image reconstruction (*e.g.*, the 2nd, 5th, and 8th columns of Figure 3 in the main paper) by using only the CAE encoder and decoder with the latent regressor skipped. After we pretrain the CAE model under two settings: w/ the alignment constraint and w/o the alignment constraint, we feed the full image (including visible patches and masked patches) to the pretrained CAE encoder and then the decoder by skipping the latent regressor, as shown in Figure 10. The pretrained CAE decoder outputs the reconstructed image. It can be seen that the image can be reconstructed with the semantics kept when training with the alignment constraint, verifying the input and the predicted representations lie in the same space. Otherwise, the reconstructed images are meaningless.

## C   CONCURRENT WORK AND MORE RESULTS

**Concurrent work.** There are concurrently-developed MIM methods, e.g., extending MIM to frequency domain (Xie et al., 2022a; Liu et al., 2022), studying the scalability of MIM (Xie et al., 2022c), combining MIM with contrastive learning (Tao et al., 2022; Jing et al., 2022; Yi et al., 2022; Huang et al., 2022b), efficient pretraining (Zhang et al., 2022; Huang et al., 2022a; Chen et al., 2022), designing mask strategy Kakogeorgiou et al. (2022); Li et al. (2022a;c), studying how MIM works (Xie et al., 2022b; Li et al., 2022b; Kong & Zhang, 2022). Other variants (Wei et al., 2022; Li et al., 2022d) extend MIM through forming the targets using semantic encoders which is essentially a supervised learning method other than self-supervised learning.

Table 7: The results of some concurrently-developed self-supervised MIM methods for semantic segmentation on ADE20K, and object detection and instance segmentation on COCO with the Cascaded Mask-RCNN framework (1× schedule). ViT-B is used for all experiments. The segmentation results of other methods are from the corresponding paper, and all the detection results are from our implementation.

| Method | Pretraining Dataset | #Epochs | ADE mIoU | COCO $AP^b$ | COCO $AP^m$ |
|---|---|---|---|---|---|
| SplitMask (El-Nouby et al., 2021) | ADE20K | 21000 | 45.7 | - | - |
| Ge$^2$-AE (Liu et al., 2022) | ImageNet-1K | 800 | 48.9 | - | - |
| A$^2$MIM (Li et al., 2022b) | ImageNet-1K | 800 | 49.0 | - | - |
| MAE (He et al., 2022) | ImageNet-1K | 1600 | 48.1 | 51.3 | 44.3 |
| iBOT (Zhou et al., 2021) | ImageNet-1K | 1600 | 50.0 | 51.2 | 44.2 |
| CAE* | ImageNet-1K | 300 | 48.3 | 51.6 | 44.6 |
| CAE* | ImageNet-1K | 800 | 49.7 | 52.8 | 45.5 |
| CAE* | ImageNet-1K | 1600 | **50.2** | **52.9** | **45.5** |

**Segmentation and detection.** Table 7 reports the results of semantic segmentation on ADE20K for some concurrent papers. We also report the results of object detection and instance segmentation under the Cascaded Mask R-CNN framework (Cai & Vasconcelos, 2021).

**Downstream classification.** We conduct fine-tuning experiments on three datasets: Food-101 (Bossard et al., 2014), Clipart (Castrejon et al., 2016) and Sketch (Castrejon et al., 2016). Results in Table 8 shows that the proposed method outperforms previous supervised method (DeiT) and self-supervised methods (DINO, MAE).

Table 8: Top-1 classification accuracy on Food-101, Clipart and Sketch. ViT-B is used here.

| Method | Supervised | Self-supervised | Food-101 | Clipart | Sketch |
|---|---|---|---|---|---|
| Random Init. | × | × | 82.77 | 52.90 | 46.42 |
| DeiT | √ | × | 91.81 | 81.18 | 73.45 |
| DINO | × | √ | 91.67 | 80.72 | 73.13 |
| MAE | × | √ | 93.19 | 80.63 | 73.87 |
| CAE* | × | √ | **93.32** | **81.84** | **74.65** |

**Impact of pretraining targets.** To study the impact of different pretraining targets on model performance, we conduct additional experiments on the RGB pixel value target. Comparing the results with DALL-E tokenizer and d-VAE tokenizer trained on ImageNet-1K, the model shows better linear probe and segmentation results but inferior in attentive probe, as shown in Table 9. Pretraining with these three targets obtains similar performance, illustrating that CAE does not rely on specific pretraining targets.

Table 9: Evaluation of different pretraining targets on the performance of CAE. ViT-B is used here. Models are trained for 1600 epochs.

| Targets | LIN | ATT | ADE Seg |
|---|---|---|---|
| DALL-E tokenizer | 70.4 | 77.1 | 50.2 |
| d-VAE tokenizer | 71.4 | 77.4 | 50.1 |
| RGB pixel value | 72.4 | 77.0 | 50.4 |

**Training costs analysis.** We report the runtime cost for the training process for ViT-B under the number of 1600 epochs in Table 10. The time cost is got with 4 machines with 8-GPU A100 and batch size is 2048. One can see that the time costs for different models are similar (MAE < CAE < BEiT slightly).

Table 10: Training cost analysis. ViT-B is used here. Models are trained for 1600 epochs.

| Method | Epochs | Total Hours | Memory/GPU |
|---|---|---|---|
| BEiT | 1600 | 115 hours | 14330 MiB |
| MAE | 1600 | 85 hours | 11222 MiB |
| CAE | 1600 | 109 hours | 13730 MiB |

