# OpenReview forum: "Context Autoencoder for Self-Supervised Representation Learning"
_ICLR.cc/2023/Conference — Submitted to ICLR 2023_

### Official Review · Reviewer_cbX1 · 2022-10-21

**Confidence:** 5
**Correctness:** 3
**Technical Novelty And Significance:** 3
**Empirical Novelty And Significance:** 3
**Recommendation:** 6

**Clarity, Quality, Novelty And Reproducibility:**

Overall, the proposed method may be of some inspiration to the community. However, some details and experiments need to be further investigated.
Lack of limitations (e.g. training costs analysis).


**Strength And Weaknesses:**

Strength:
Good results on downstream transferring learning, especially for semantic segmentation.
Clear idea, decoupling the space of encoder and decoder explicitly may benefit the learning of representation.

Weaknesses:
(1) In fact, the “Context” in the Title does not correspond well with the interpretation in the main manuscript. MAE, BEiT and other Masked Image Modeling approaches can also be understood as including context information. I am not sure what specific meaning the context in the paper refers to.
(2) Several details and designs are not clear:
-- Why is the masking rate set to 50%? (Section 3.2) Why not 75% or other values?
-- How to balance the two loss functions? (Eq (1))
-- How to choose the number of layers in the Latent Contextual Regressor and Decoder? The number of layers may be related to the performance.

(3) Questions about the experiments:
-- In fact, the finetuning results in Table 1 are only slightly higher than MAE (0.3%).
-- How about removing the Decoder? That’s only containing the latent contextual regressor. In addition, in Table 2, why not use the finetuning or linear probe metrics?
-- In Table 3, according to the MAE paper, the detection performance of MAE with ViT-B is 50.3%, not 48.4%. In addition, MAE+ViTDet [1] has already achieved higher results (ViT-B: 51.6% and ViT-L: 56.7%).
[1] Yanghao Li, et al. Exploring Plain Vision Transformer Backbones for Object Detection. ECCV2022.


**Summary Of The Paper:**

The paper proposes to decouple the encoder and decoder space of MAE, thus improving the representation quality. Similar to contrastive learning, a Latent contextual regressor is further used to align the predicted masked patch with the encoded masked patch. Experiments on downstream object detection and semantic segmentation show the effectiveness of the proposed method.

**Summary Of The Review:**

I am inclined to accept the paper if the author can dispel my concerns well.

---

> ### Author Response · Authors · 2022-11-18
> **Response to Reviewer cbX1 (Part 2/2)**
>
> #### **Q7: [In Table 2, why not use the finetuning or linear probe metrics?]**
>
> A: The fine-tuning and linear probing results are given as follows. These two metrics are consistent with those metrics in Table 2. For table clarity, we did not show these two metrics in Table 2 in the submitted paper.
>
> | Decoder | Align | FT   | LIN  |
> | ------- | ----- | ---- | ---- |
> | &#10006;       | &#10006;     | 82.9 | 60.3 |
> | &#10004;       | &#10006;     | 83.4 | 63.1 |
> | &#10004;       | &#10004;     | 83.6 | 64.1 |
>
>
>
>
>
> #### **Q8: [In Table 3, according to the MAE paper, the detection performance of MAE with ViT-B is 50.3%, not 48.4%. In addition, MAE+ViTDet [1] has already achieved higher results (ViT-B: 51.6% and ViT-L: 56.7%).]**
>
> A: The experiment settings are different.
>
> In Table 3, the detection results are reported from Mask R-CNN of 12 epochs (1$\times$ schedule). In the MAE paper, the results are from Mask R-CNN with 100 epochs. When they train for 25 epochs (please see Figure 2 in [2] from some MAE authors), the detection performance drops to ~48.1%, which is slightly lower than our implementation (48.4) under 12 epoch training.
>
>
> In [1], the method, MAE + ViTDet, is different from ours, and is not comparable to our result. The results in [1] are from 100 epochs and new feature pyramid neck, large-scale jitter, and other tricks.
>
> [1] Exploring Plain Vision Transformer Backbones for Object Detection. \
> [2] Benchmarking Detection Transfer Learning with Vision Transformers.
>
>
>
> #### **Q9: [Training costs analysis.]**
>
> A: We report the runtime cost for the training process for ViT-B under the number of 1600 epochs. The time cost is got with 4 machines with 8-GPU A100 and batch size is 2048. One can see that the time costs for different models are similar (MAE < CAE < BEiT slightly).
>
> | Method | Epochs | Total Hours | Memory/GPU |
> | ------ | ------ | ------------ | ---------- |
> | BEiT   | 1600   | 115 hours         | 14330 MiB  |
> | MAE    | 1600   | 85 hours        | 11222 MiB  |
> | CAE    | 1600   | 109 hours         | 13730 MiB  |
>
>
> *We sincerely appreciate your comments. Please feel free to let us know if you have further questions.*
>
> Best,  \
> Authors

---

> ### Author Response · Authors · 2022-11-18
> **Response to Reviewer cbX1 (Part 1/2)**
>
> Thank you for your insightful and constructive comments! We have added additional experiments and modified our paper according to your comments.
>
> #### **Q1: [In fact, the “Context” in the Title does not correspond well with the interpretation in the main manuscript. MAE, BEiT and other Masked Image Modeling approaches can also be understood as including context information. I am not sure what specific meaning the context in the paper refers to.]**
>
> A: Thanks a lot. We use "context" by considering two aspects:
> 1. The regressor is to predict the representations of masked patches from the context, visible patches.
> 2. The inputs of the encoder are about visible patches, the inputs of the decoder are about masked patches, the two inputs are mutual contexts.
>
>
>
> #### **Q2: [Why is the masking rate set to 50%? (Section 3.2) Why not 75% or other values?]**
>
> A: Good point. We used to test the results for masking ratios 40%, 50%. We found that ratio 0.5 gets better results than ratio 0.4. During rebuttal, we followed your advice and tested another value 60%, which improves the performance of linear probing and attentive probing, while the semantic segmentation performance is reduced by 0.2 than that of mask ratio 0.5.
>
> | Mask Ratio | Linear Probing | Attentive Probing | Semantic Segmentation |
> | ---------- | -------------- | ----------------- | --------------------- |
> | 0.4        | 63.1           | 73.0              | 47.2                  |
> | 0.5        | 64.1           | 73.8              | 48.3                  |
> | 0.6        | 64.8           | 74.2              | 48.1                  |
>
>
>
> #### **Q3: [How to balance the two loss functions? (Eq (1))]**
>
> A: We discussed it in "hyperparameter choice" in **Appendix** of the submitted paper. We tried three choices,  $\lambda$ = 1,  $\lambda$ = 1.5 and $\lambda$ = 2. The linear probing results are 63.4, 63.7, and 64.1, thus we choose $\lambda$ = 2.
>
>
> #### **Q4: [How to choose the number of layers in the Latent Contextual Regressor and Decoder? The number of layers may be related to the performance.]**
>
> A: We tried four choices: 1 layer, 2 layers, 4 layers, and 5 layers (4 layers means both the number of layers in Latent Contextual Regressor and Decoder are 4). We empirically observed that 4 layers perform overall the best.
>
> | Layer Num | Linear Probing | Attentive Probing |
> | --------- | -------------- | ----------------- |
> | 1         |     58.7           |  67.5                 |
> | 2         |      62.1          |            71.7        |
> | 4         | 64.1           | 73.8              |
> | 5         | 64.2           | 73.7              |
>
>
>
>
> #### **Q5: [In fact, the finetuning results in Table 1 are only slightly higher than MAE (0.3%).]**
>
> A: Thanks.
>
> The results are in line with our expectations: The fine-tuning performance for the MIM methods and contrastive learning are similar. In Section 5.2 of the submitted paper, we presented this discussion: "The fine-tuning performance for these methods are very similar and there is only a minor difference similar to the observation (Zhou et al., 2021). We think that the reason is that self-supervised pretraining and fine-tuning are conducted on the same dataset and no extra knowledge is introduced for image classification. The minor difference might come from the optimization aspect: different initialization (provided by pretrained models) for fine-tuning." It is also debatable whether fine-tuning performance is proper for pretrained model evaluation.
>
>
> #### **Q6: [How about removing the Decoder? That’s only containing the latent contextual regressor.]**
>
> A: This is a good point.
>
> In Table 2, we reported the result with the decoder removed and only using a linear layer following the regressor (without alignment) to reconstruct the target: 60.3, 71.2, 47.0 (linear, attentive, ADE seg) vs. 63.1, 72.7, 47.1 for decoder kept and alignment removed.
>
> During rebuttal, we conducted an experiment: with the decoder removed and alignment kept. The results are 62.0, 71.5, 47.1. The performance drops compared with our full model (64.1, 73.8, 48.3), illustrating the usefulness of the decoder. Without the decoder, the reconstruction target contradicts the alignment constraint.

---

> ### Author Response · Authors · 2022-12-03
> **Looking forward to post-rebuttal discussions**
>
> Dear Reviewer #cbX1,
>
> We sincerely appreciate your time and efforts in reviewing our paper, which would help us improve our final paper!
>
> As the deadline for discussion is approaching, we are happy to provide any additional clarifications that you may need. In our previous response, we have carefully studied your comments and made detailed responses summarized below:
>
> - We interpreted the word 'context'.
> - We added ablation studies on mask ratio.
> - We detailed our hyper-parameter selection settings.
> - We added ablative experiments on the number of layers.
> - We made detailed explanations about the experimental metrics and results for finetuning and linear probing.
> - We conducted new experiments about removing the decoder.
> - We provided training costs analysis.
>
> We hope the above experiments and analyses could clarify your concerns. Please don’t hesitate to let us know if there are any additional clarifications we can offer. Thanks for your time and efforts!
>
> Best, \
> Authors

---

> > ### Comment · Reviewer_cbX1 · 2022-12-07
> > **Comments after rebuttal**
> >
> > Thanks for your thorough response. I am inclined to accept the paper.

---

### Official Review · Reviewer_b4Df · 2022-10-23

**Confidence:** 4
**Clarity, Quality, Novelty And Reproducibility:** Please see Strength And Weaknesses se…
**Correctness:** 3
**Technical Novelty And Significance:** 2
**Empirical Novelty And Significance:** 3
**Recommendation:** 5

**Strength And Weaknesses:**

** Strengths
- Comparisons to the concurrent methods with analysis and discussion
- Extensive experimental results achieving state-of-the-art performances on various downstream tasks
- Clear implementation details
- Using pretrained tokenizer for masking strategy is interesting

** Weaknesses
- The idea of reconstructing just the features of the masked patches is also used in iBOT
- The reliance on a pre-trained tokenizer could be cumbersome. While being with 250M images as extra data, there are not many insights and improvements compared to previous works, e.g. competitive performances compared to iBOT and MAE even they don't require using such pre-trained tokenizers
-

**Summary Of The Paper:**

The authors proposed a method for self-supervised representation learning based on masked image modeling. Unlike previous approaches, the proposed method decouple representation learning part and pretext task completion part where the learning signal comes from the reconstruction in the encoded representation space rather than in image space. Alignment constraint is introduced encouraging the predicted representations to be lied in the encoded one. The experiments are conducted on three different downstream tasks; semantic segmentation, object detection, and instance segmentation, surpassing the previous approaches.

**Summary Of The Review:**

As the weaknesses outweigh the strengths, I lean towards 5: marginally below the acceptance threshold at this point.

---

> ### Author Response · Authors · 2022-11-18
> **Response to Reviewer b4Df**
>
>
> Thank you for your insightful and constructive comments! We have added additional experiments and modified our paper according to your comments.
>
> #### **Q1: [The idea of reconstructing just the features of the masked patches is also used in iBOT.]**
>
> A: Yes. Our approach and iBOT both predict the representations of the masked patches: alignment in CAE, and distillation for in-view patch tokens.
>
> There are significant differences in how to compute the target representations and how to predict the representations.
> (1) In CAE, the representation for the alignment target is the representation computed by feeding the masked patches into the encoder, and the predicted features computed from the regressor with the representations of visible patches computed from the encoder as the input: masked patches $\rightarrow$ encoder $\rightarrow$ target representations of masked patches; visible patches $\rightarrow$ encoder $\rightarrow$ representations of visible patches $\rightarrow$ regressor $\rightarrow$ predicted representations of masked patches.
>
> (2) In contrast, for iBOT, both views, view with masked tokens and full view, are sent to the encoder and projector, and EMA encoder and EMA projector, generating predicted representations of masked patches, and target representations of masked patches.
>
> (3) In addition, iBOT follows DINO to center and sharpen the representations for formulating the loss, which helps avoid collapse. CAE, instead, directly computes the $\ell_2$ loss.
>
>
>
> #### **Q2: [The reliance on a pre-trained tokenizer could be cumbersome. While being with 250M images as extra data, there are not many insights and improvements compared to previous works, e.g. competitive performances compared to iBOT and MAE even they don't require using such pre-trained tokenizers]**
>
> A: Thanks. We agree with you and also had the concern about that the DALL-E tokenizer (pretrained on 250 M images) might be a problem. In the submitted paper, we thus reported the results with the tokenizer trained on ImageNet-1K in Table 1, Table 2, Table 4 (termed as "CAE" without * ). The results imply that the two tokenizers achieve similar results. This means that the competitive performance does not come from the DALL-E tokenizer. And we also are working on training the CAE using the pixel RGB formulating the reconstruction target instead of the pretrained tokenizer, and currently the results on ViT-base are: 72.36, 77.04, 83.89, and 50.40 (linear probing, attentive probing, finetuning, and ADE segmentation), similar to the results with tokenizer, 70.40, 77.10, 83.90, 50.20.
>
>
> *Please let us know if you have any further questions about our paper. We really appreciate your time! Thank you!*
>
> Best, \
> Authors

---

> ### Author Response · Authors · 2022-12-03
> **Looking forward to post-rebuttal discussions**
>
> Dear Reviewer #b4Df,
>
> We sincerely appreciate your time and efforts in reviewing our paper, which would help us improve our final paper!
>
> As the deadline for discussion is approaching, we are happy to provide any additional clarifications that you may need. In our previous response, we have carefully studied your comments and made detailed responses summarized below:
>
> - We reiterated our contribution and detailed the main differences with iBOT.
> - We highlighted our results without using the DALL-E tokenizer.
>
> Please don’t hesitate to let us know if there are any additional clarifications we can offer. Thanks for your time again!
>
> Best, \
> Authors

---

### Official Review · Reviewer_TXRx · 2022-10-24

**Confidence:** 4
**Correctness:** 3
**Technical Novelty And Significance:** 2
**Empirical Novelty And Significance:** 2
**Recommendation:** 6

**Clarity, Quality, Novelty And Reproducibility:**

In the paragraph of `Relation to BEiT and MAE.`, the authors mentioned that `In MAE (He et al., 2022), the so-called
decoder may play a partial role for representation learning as the representations of the visible
patches are also updated in the MAE decoder`, which is somewhat inaccurate and makes me confused as the "updated representation" for visible patches is neither supervised with the reconstruction loss nor used for downstream tasks. Whether using extra layers to process representations for visible patches should not be a key difference and the cross attention used in this paper also has MLP for the key/value (which are the visible patches feature).

Figure 3 is somewhat hard to follow without referring to the caption, might be better to add some illustration about how the output is generated.

Misc: I believe the citation of LARS is incorrect.

**Strength And Weaknesses:**

The paper is well-written. The motivation of this paper is clear and easy to follow.

The proposed attentive probing is interesting and well-motivated, however, it seems that even with this metric the CAE trained with 800 epochs using ViT-B cannot surpass the contrastive learning-based model MoCo-v3 according to Table 1.

Performance wise, the downstream task results on COCO look superior, especially when using ViT-B, but when moving to ViT-L the gain seems diminished (from ~2 points to ~0.5 points compared with MAE on 1600ep).

The technical part seems to be somewhat incremental, how important/critical the alignment is? Seems like this issue only happens when making predictions in the representation space. For example, MAE without the alignment loss will not show meaningless output like Figure 3. The role of the cross-attention based latent contextual regressor is also not clear, one could also use the output from the MAE decoder's first layer to compute the alignment loss.

**Summary Of The Paper:**

This paper presents context autoencoder (CAE), which follows BEiT and uses an additional latent contextual regressor to make predictions for the masked patches with cross attention over the visible patches. By doing that we can encourage the masked patches' representation predicted from visible patches are aligned with the one from the encoder and then benefits the learning and downstream transferring. Experiments on downstream tasks also show that it can achieve better performance compared with the baseline BEiT and MAE.

**Summary Of The Review:**

I think this paper shows some interesting findings and good performance, although the technical contribution is somewhat limited from my perspective. Thus my rating is 6.

---

> ### Author Response · Authors · 2022-11-18
> **Response to Reviewer TXRx (Part 2/2)**
>
>
> #### **Q4: [Figure 3 is somewhat hard to follow without referring to the caption, might be better to add some illustration about how the output is generated.]**
>
> A: Thanks a lot! We have followed your advice and added more illustrations in Appendix to make it clearer.
>
>
> #### **Q5: [Misc: I believe the citation of LARS is incorrect.]**
>
> A: Thanks a lot! We followed and fixed it in the revised paper.
>
>
> *We sincerely appreciate your comments. Please feel free to let us know if you have further questions.*
>
> Best, \
> Authors

---

> ### Author Response · Authors · 2022-11-18
> **Response to Reviewer TXRx (Part 1/2)**
>
>
> We appreciate the positive and insightful comments from you! We address your concerns in detail below.
>
> #### **Q1: [The proposed attentive probing is interesting and well-motivated, however, it seems that even with this metric the CAE trained with 800 epochs using ViT-B cannot surpass the contrastive learning-based model MoCo-v3 according to Table 1.]**
>
> A: Thanks a lot for pointing this out.
>
> Attentive probing aims to select the object region information for better evaluating the MIM pretrained encoder, and the probing quality still depends on the representations of the pretrained encoder.
>
> Like other MIM methods, CAE and MAE do not have the advantage over contrastive learning methods (MoCo v3 and DINO) in terms of linear probing and attentive probing, which is because the pretraining of contrastive learning is relatively easier than existing MIM methods as contrastive learning mainly cares about the 1000 classes and MIM methods may care about the classes beyond the 1000 classes.
>
> #### **Q2: [The technical part seems to be somewhat incremental, how important/critical the alignment is? Seems like this issue only happens when making predictions in the representation space. For example, MAE without the alignment loss will not show meaningless output like Figure 3. The role of the cross-attention based latent contextual regressor is also not clear, one could also use the output from the MAE decoder's first layer to compute the alignment loss.]**
>
> A: This is a good point. Regarding the importance of alignment, we studied its effect on our CAE architecture in Table 2, showing it improves the gains of 1.1%, 1.2%, 1.2% for attentive probing, ADE segmentation, and COCO object detection, and validated that alignment is helpful for ensuring the predictions are made in the encoder representation space (evidenced in Figure 3).
>
> Regarding alignment for MAE, we agree that "MAE without the alignment loss will not show meaningless output like Figure 3". The main reason is that we removed the regressor after training for reconstructing the image in Figure 3. If we do not remove the regressor, CAE without alignment will also not show meaningless output.
>
> We followed your advice and conducted an initial experiment: "use the output from the MAE decoder's first layer to compute the alignment loss". The results of linear probing and attentive probing are 61.5% and 71.2%, respectively. This is almost the same as the baseline (61.5%, 71.1%).
>
>
> #### **Q3: [In the paragraph of "Relation to BEiT and MAE", the authors mentioned that "In MAE (He et al., 2022), the so-called decoder may play a partial role for representation learning as the representations of the visible patches are also updated in the MAE decoder", which is somewhat inaccurate and makes me confused as the "updated representation" for visible patches is neither supervised with the reconstruction loss nor used for downstream tasks. Whether using extra layers to process representations for visible patches should not be a key difference and the cross attention used in this paper also has MLP for the key/value (which are the visible patches feature).]**
>
> A: Thanks a lot! Our clarification is as follows.
>
> For self-supervised MIM methods, the basic goal is to learn an encoder for representation learning through completing the pretext task, masked patch reconstruction. We hypothesize that pretext task (masked patch reconstruction) completion needs a semantically-good representation, and we hope that representation learning lies in the encoder that will be used for downstream tasks.
>
> In our CAE, the representation of visible patches is only used as the key/value (with a linear projection layer) of the cross-attention in the latent contextual regressor rather than the decoder. In a model inference pass, the encoded representation is fixed after the encoder and expected to estimate the representation of masked patches through mask queries.
>
> In MAE, the updated representation of visible patches in the decoder might further improve the semantics of visible patches that helps better on masked patch prediction, and accordingly, better minimize the reconstruction loss of masked patches. The updated representations in the decoder are not included in the encoder, and might benefit downstream tasks but are not leveraged by downstream tasks.
>
> Your comment "the cross attention used in this paper also has MLP for the key/value (which are the visible patches feature)" is correct, and the linear projection layer may update the representation of visible patches a little. This is exactly the reason that we need an alignment constraint for encouraging the input and output of the cross-attention based regressor to lie in the same encoded representation space and eliminating the effect of MLP in the cross-attention.

---

### Official Review · Reviewer_KNoM · 2022-11-01

**Confidence:** 4
**Correctness:** 3
**Technical Novelty And Significance:** 3
**Empirical Novelty And Significance:** 3
**Recommendation:** 6

**Clarity, Quality, Novelty And Reproducibility:**

The paper is well written with most of the details clearly presented.

The paper’s quality is good in general but missed some technical details.

The paper is an improvement of BEIT and MAE. The novelty is limited as the comments above.


**Strength And Weaknesses:**

Strength
1. The paper is well written and easy to follow. The comprehensive analysis between the proposed approach and the existing SSL works add a valuable insight into its contributions.
2. The evaluation confirms the proposed approach can outperform the SOTA SSL approach on various backbone and downstream tasks.

Weakness
1. The novelty of this paper is limited. The proposed CAE is an improvement work over MAE and BEIT by further exploiting the knowledge between the masked and visible patches in the latent space. There is no sufficient evidence that the proposed cross-attention module can further improve representation performance learned by CAE.
2. The proposed CAE approach used a random block-wise masking approach. It is necessary to add an ablation study about the performance between various masking approaches, e.g. random patch masking. Moreover, the masking ratio is set to 0.5 by default. What would the performance be for other masking ratios? An ablation study would help clarify that.
3. Some technical details are missed in the paper. E.g. how to decode \bar{Y}_m from \bar{Z}_m in figure 7a is not clear.


**Summary Of The Paper:**

The paper proposed a novel contextual autoencoder framework for SSL. The proposed CAE approach introduced a cross-attention module to learn the latent space knowledge between the visible and masked patches. The paper had a comprehensive analysis between the proposed approach and the existing SSL approaches which gave some insight to the reader to understand the core contributions.


**Summary Of The Review:**

The paper is an improvement work over BEIT and MAE with some minor novelties. The proposed CAE algorithm achieved the SOTA performance on various backbone and downstream tasks. The paper can be improved by clarifying the technical details and adding ablation studies.

---

> ### Author Response · Authors · 2022-11-18
> **Response to Reviewer KNoM**
>
>
> We appreciate the positive and insightful comments from you! We address your concerns in detail below.
>
> #### **Q1: [The novelty of this paper is limited. The proposed CAE is an improvement work over MAE and BEIT by further exploiting the knowledge between the masked and visible patches in the latent space. There is no sufficient evidence that the proposed cross-attention module can further improve representation performance learned by CAE.]**
>
> A: We would like to clarify the main point of this paper: make predictions for masked patches in the encoded latent representation space through the alignment constraint. And the cross-attention is one implementation for the contextual regression module that outputs the predicted representations of masked patches for alignment. The ablation study result in Table 2 shows the effect of alignment; and the results in Table 1, Table 3, and Table 4 demonstrate the effectiveness of the whole CAE architecture design.
>
> #### **Q2: [The proposed CAE approach used a random block-wise masking approach. It is necessary to add an ablation study about the performance between various masking approaches, e.g. random patch masking. Moreover, the masking ratio is set to 0.5 by default. What would the performance be for other masking ratios? An ablation study would help clarify that.]**
>
> A: This is a good point. Our CAE implementation follows the BEiT implementation. Similar to the observations from BEiT, block-wise masking performs better than random patch masking according to our early studies. The performance of BEiT semantic segmentation on ADE20K drops from 44.65% to 42.93%, according to Table 4 in the BEiT paper.
>
> Regarding the block masking ratio, we did some experiments for ratios 0.4 and 0.5, and found that ratio 0.5 gets better results than ratio 0.4, so we chose 0.5 for our work.
>
> During rebuttal, we followed the suggestion and did experiments for a higher mask ratio (0.6), which further improves the performance of linear probing and attentive probing, while the semantic segmentation performance is reduced by 0.2%.
>
>
> | Mask Ratio | Linear Probing | Attentive Probing | Semantic Segmentation |
> | ---------- | -------------- | ----------------- | --------------------- |
> | 0.4        | 63.1           | 73.0              | 47.2                  |
> | 0.5        | 64.1           | 73.8              | 48.3                  |
> | 0.6        | 64.8           | 74.2              | 48.1                  |
>
>
>
> **Q3: [Some technical details are missed in the paper. E.g. how to decode $\bar{\mathbf{Y}}_m$ from $\bar{\mathbf{Z}}_m$ in figure 7a is not clear.]**
>
> A: Thanks a lot for raising this point. We need to make the description clearer. (1) We do not decode $\bar{\mathbf{Y}}_m$ from $\bar{\mathbf{Z}}_m$. (2) $\bar{\mathbf{Y}}_m$ is the reconstruction target (e.g., in the tokenizer form), and used to compute the loss between prediction $\mathbf{Y}_m$ and target $\bar{\mathbf{Y}}_m$. (3) $\bar{\mathbf{Z}}_m$ is the representation of masked patches, computed through the encoder with masked patches as input (See more in Figure 1).
>
> *We sincerely appreciate your comments. Please feel free to let us know if you have further questions.*
>
> Best, \
> Authors

---

### Author Response · Authors · 2022-11-18
**General Response: Contributions and New Experiments**



We sincerely appreciate all reviewers’ time and efforts in reviewing our paper. We are glad to find that reviewers generally recognized our contributions:
* **Model.** valuable insight [KNoM]; interesting and well-motivated [TXRx, b4DF]; clear idea [cbX1];
* **Experiments.** comprehensive analysis, extensive experiments [KNoM, b4DF]; outperform the SOTA SSL approach [KNoM, b4DF, cbX1];
* **Writing.** well written and easy to follow [KNoM, TXRx]; clear implementation details [KNoM, b4DF];


And we also thank all reviewers for their insightful and constructive suggestions, which help a lot in further improving our paper. In addition to the pointwise responses below, we summarize supporting experiments added in the rebuttal according to reviewers’ suggestions.

**New Experiments**
* Ablation studies on mask ratio [KNoM, cbX1].
* Use the output from the MAE decoder's first layer to compute the alignment loss [TXRx].
* Ablative experiments on the number of layers [cbX1].
* Experiments about removing the decoder [cbX1].
* Linear and finetune results in our ablation studies [cbX1].
* Training costs analysis [cbX1].


All additional experiments and modifications have been delivered in our revision (appendix). We hope our pointwise responses below could clarify all reviewers’ confusion and alleviate all concerns. We thank all reviewers’ time again.

Best, \
Authors

---

### Decision · Program_Chairs · 2023-01-20

**Decision:**

Reject

**Justification For Why Not Higher Score:**

The paper offers limited technical novelty over prior methods (MAE, BEiT, iBOT) and also gives relatively minor performance gains over these methods.

**Justification For Why Not Lower Score:**

N/A

**Metareview: Summary, Strengths And Weaknesses:**

The paper proposes the Contextual Autoencoder (CAE) for self-supervised learning. It builds on MAE and BEiT to perform masked image modeling with Vision Transformers (ViTs). The primary technical novelty compared to these prior works is that CAE attempts to predict the latent representations of the masked patches using a latent content regressor module in addition to predicting the full image representation from the decoder; this essentially amounts to training MAE with dVAE targets, and an additional alignment loss on latent representations in the decoder.

Initial reviews for this paper were borderline. Reviewers found the paper well-written and easy to follow, understood the key difference between CAE and prior work, and agreed that CAE can outperform prior SSL methods in some contexts. Reviewer sentiment was slightly positive overall, but no reviewers argued strongly in favor of accepting the paper. Reviewers asked for many clarifications and extra experiments which the authors supplied in their responses, including results with different mask ratios (KNoM and cbX1), clarifications on Figure 7a (KNoM), results on using the MAE decoder’s first layer representations to compute the alignment loss (cbX1), experiments removing the decoder (cbX1), linear and finetuning results for the ablation study (cbX1), and a discussion of training costs (cbX1). While the reviewers (aside from cbX1) did not meaningfully engage with these extensive author responses, in the AC’s view the additional results and explanations supplied by the authors were successful in addressing the many of the specific issues raised by reviewers.

However, in the ACs view there is one key issue that was not fully addressed by the author's responses: novelty and significance with respect to MAE and BEiT. A common theme among the reviewers is that the proposed method is a small variation on MAE without significant technical novelty (KNoM: “The novelty of this paper is limited.”; TXRx: “the technical contribution is somewhat limited”; b4Df: “there are not many insights and improvements compared to previous works”). The proposed method is not exactly the same as any presented in prior work, but in light of MAE, BEiT, and iBOT there are no “big ideas” or “big insights” that are significantly different from the body of prior work in this area.

In the ACs view, minor technical novelty is not necessarily a reason to reject the paper; simple changes to existing methods can be of benefit to the community if those changes yield significant experimental results. While the authors have performed an extensive and admirable set of ablations and experiments, taken as a whole the experiments do not demonstrate that CAE has a definitive and consistent advantage over prior methods. On ImageNet (Table 1), CAE vs MAE gives just +0.3 for finetuning on both ViT-B and ViT-L; while CAE gives significant gains over MAE for linear and attention probing, it is outperformed by MoCo-v3 and iBOT under those evaluation protocols. For semantic segmentation (Table 3), CAE gives +0.1 vs iBOT for ViT-B, and +1.0 vs MAE for ViT-L. For object detection, CAE vs MAE gives larger improvements on both detection (+1.8 AP) and instance segmentation (+1.6 AP) on ViT-B, but these advantages shrink on ViT-L (+0.6 and +0.5). As a whole, these results show CAE as having some advantages over prior methods, but other than COCO detection with ViT-B and ADE20K segmentation with ViT-L, CAE does not offer large performance improvements over prior methods; it also uses slightly more resources (time and memory) vs MAE for training.

The AC met with Reviewer KNoM to discuss this paper, who largely agreed with the points above. Overall, the experimental results are not significant enough to offset the relatively minor technical novelty presented by the paper and so the paper is not suitable for publication at this time.

**Summary Of Ac-Reviewer Meeting:**

The AC met with Reviewer KNoM to discuss this paper. The main points of discussion were the minor technical novelty of the paper compared to prior work, and the mixed experimental results. Overall we felt that the results were not significant enough to outweigh the minor technical novelty.